# Antiviral Peptides as Anti-Influenza Agents

**DOI:** 10.3390/ijms231911433

**Published:** 2022-09-28

**Authors:** Mariangela Agamennone, Marialuigia Fantacuzzi, Giovanni Vivenzio, Maria Carmina Scala, Pietro Campiglia, Fabiana Superti, Marina Sala

**Affiliations:** 1Department of Pharmacy, University “G. d’Annunzio” of Chieti-Pescara, Via dei Vestini 31, 66100 Chieti, Italy; 2Department of Pharmacy, University of Salerno, Via Giovanni Paolo II 132, 84084 Fisciano, Italy; 3National Centre for Innovative Technologies in Public Health, National Institute of Health, Viale Regina Elena 299, 00161 Rome, Italy

**Keywords:** Influenza A virus, peptides, drugs, hemagglutinin, neuraminidase, fusion peptide, amphiphilic peptides

## Abstract

Influenza viruses represent a leading cause of high morbidity and mortality worldwide. Approaches for fighting flu are seasonal vaccines and some antiviral drugs. The development of the seasonal flu vaccine requires a great deal of effort, as careful studies are needed to select the strains to be included in each year’s vaccine. Antiviral drugs available against Influenza virus infections have certain limitations due to the increased resistance rate and negative side effects. The highly mutative nature of these viruses leads to the emergence of new antigenic variants, against which the urgent development of new approaches for antiviral therapy is needed. Among these approaches, one of the emerging new fields of “peptide-based therapies” against Influenza viruses is being explored and looks promising. This review describes the recent findings on the antiviral activity, mechanism of action and therapeutic capability of antiviral peptides that bind HA, NA, PB1, and M2 as a means of countering Influenza virus infection.

## 1. Introduction

### 1.1. Influenza Virus

Influenza is an acute respiratory disease associated with high morbidity, mortality, and significant economic losses worldwide [1]. In humans, Influenza viruses are mainly transmitted through respiratory droplets, including aerosols, which are expelled when an infected person coughs, sneezes, breathes, talks, or sings. Transmission through intermediate objects and surfaces contaminated by the virus can also occur. Flu symptoms are fever or a feeling of fever, runny and stuffy nose, cough, sore throat, headaches, fatigue, tiredness, and muscle or body aches [2]. Typically, infected people recover within a few days or, at most, 1 to 2 weeks. However, some individuals, such as adults older than age 65, young children under age 5, and people with certain health issues like a weakened immune system or chronic diseases, are at risk of aggravating their health conditions resulting in hospitalization and even death [3]. The Influenza A virus (IAV) and the Influenza B virus (IBV) are the two primary human Influenza viruses that cause seasonal flu epidemics each year. IAV and IBV can evolve to escape pre-existing immunity and gain competitive benefit through mutations in surface proteins which lead to new antigenic variants [4]. Natural selection acts worldwide through the combination of virus evolution, epidemiology, and human behavior [5]. Therefore, new viral strains have appeared over the centuries, producing human pandemics, and causing widespread disease and death. Notably, four worldwide (pandemic) Influenza outbreaks have occurred in just over a hundred years. In the 20th century, three Influenza pandemics, named by their presumed places of origin as Spanish, Asian, and Hong Kong flu, occurred in 1918, 1957, and 1968, respectively [6]. While, as already reported, IAV and IBV are responsible for the annual Influenza epidemics in humans, only IAV is capable of causing Influenza pandemics, as IBV does not have this potential because it does not have an animal reservoir [7]. Specifically, an H1N1 IAV strain was responsible for the 1918 pandemic, during which it infected about 500 million people and killed 50 million [8]. In the case of the two pandemics of 1957 and 1968, the hemagglutinin (HA) of the causative viruses showed considerable variations compared to immediately antecedent strains: the Asian IAV responsible for the 1957 pandemic has been identified as H2N2, whereas an H3N2 IAV strain was responsible for the 1968 pandemic [6]. In March and April 2009, a new H1N1 IAV (formerly known as swine flu) was isolated in Mexico and California, respectively [9], and, spreading out rapidly worldwide, affected over 214 countries, causing over 18,449 deaths [10], and resulting in the first pandemic of the 21st century. By the end of April, the World Health Organization (WHO) stated a phase-5 pandemic and, on 11 June 2009, declared a phase-6 pandemic level. The new virus has been called A/(H1N1) pdm09 [11,12].

In 2020, the dramatic emergence of severe acute respiratory syndrome coronavirus 2 (SARS-CoV-2) affected flu surveillance and negatively impacted reporting of new cases and the characterization of circulating Influenza viral strains [13,14]. However, in the WHO European region, it has been reported that non-pharmaceutical COVID-19 interventions slowed the circulation of Influenza in the 2021/22 season but also led to the selection of new variants. In particular, numerous viral strains have been identified as antigenically different from the vaccine components [13]. Another important observation about the impact of IAV infection in the “COVID-19 era” is that superinfection with IAV in patients with COVID-19 causes more severe diseases [15,16,17,18,19,20,21,22,23] and can double the risk of death [24]. Indeed, it has been observed that patients co-infected with SARS-CoV-2 and IAV were approximately twice as likely to die compared to patients with SARS-CoV-2 alone [25]. Therefore, the co-circulation of IAV and SARS-CoV-2 is an additional major threat to public health. These observations underline the importance of the study of preventive and therapeutic strategies to fight flu by developing vaccines for prophylaxis and identifying antivirals for the treatment of patients.

IAV is a single-stranded RNA enveloped virus with a genome consisting of 8 negative-sense segments encoding 11 proteins [26] (Table 1).

The pleomorphic viral particles can be spherical or filamentous, with a diameter of about 120 nm. Filamentous particles can reach a length greater than 300 nm (Figure 1).

Virions consist of envelope, M1, and core. The viral envelope is a lipid bilayer containing three membrane proteins: HA, NA, and the proton channel M2 [27] (Figure 2).

HA and NA are embedded in the lipid bilayer, protrude on the viral surface as “spikes”, and represent the two major viral antigenic determinants. IAVs are divided into subtypes based on the genetic and antigenic properties of HA and NA and their combinations. Currently, 18 HA and 11 NA subtypes of IAVs have been identified [28]. Changes in both types of “spikes” result in the emergence of antigenically new strains that escape host immunity, causing reinfections.

The mechanism of internalization of IAV into susceptible cells has been the subject of numerous studies since the early stages of infection are ideal targets for intervention strategies. HA, composed of a trimer where each monomer is formed by a globular head domain (HA1) and a stalk domain (HA2), is the glycoprotein projection responsible for virus attachment and internalization into host cells. The first phase is initiated by the interaction of HA with galactose-bound sialic acid on cell plasma membranes [29]. Mammalian IAV HA proteins predominantly bind to sialic acid with an α2,6 bond to the penultimate sugar, while avian IAVs bind to α2,3 bound sialic acid [30,31,32,33]. Recently, however, a sialic acid-independent attachment process and a possible role of NA in IAV internalization into the host cell have been described [34]. IAV enters susceptible cells via clathrin-mediated endocytosis, clathrin, and caveolae-independent endocytosis [35], or micropinocytosis [36]. The viral particles are then internalized in the endosome, where they encounter acid pH conditions. As a matter of fact, within the late endosome, the environment has a slightly acidic pH [37], and protonation induces a significant conformational change in HA. This triggers the fusion of viral and endosomal membranes and causes the release of viral ribonucleoproteins (vRNPs) into the cytoplasm [38].

The process takes place through several stages. In conditions of pH around 6.0, the M1 protein undergoes a conformational change; it dissociates from the vRNP [39,40,41], while the M1-lipid interactions do not undergo variations [42]. At pH 5.5, HA conformational change occurs, leading to contact between the viral and endosomal membranes and the formation of the small fusion pore [43,44]. A further decrease in pH leads to disassembly of the M1 scaffold polymer, widening the fusion pores, and entering vRNP into the cytoplasm [45]. Hence, pH-dependent disintegration of the M1 protein scaffold is an essential prerequisite for efficient IAV infection [42]. The vRNP is then transferred to the nucleus. Unlike the virus internalization process, the trafficking of vRNP towards the nucleus is highly dependent on the host cell’s machinery [46]. Inside the nucleus, the RNA-dependent RNA polymerase carries out genome transcription and replication [47]. IAV protein synthesis depends on the host cell translation mechanism. After leaving the nucleus, the translation of viral mRNAs for PB1, PB2, PA, NP, NS1, NS2, and M takes place through cytosolic ribosomes, while that of mRNAs for HA, NA, and M2 is dependent on the ribosomes associated with the endoplasmic reticulum [48]. The viral particle assembly and budding occur at the plasma membrane’s level.

### 1.2. Treatment of Flu

The anti-Influenza virus strategy includes the use of vaccines and antiviral drugs. However, the clinical use of these therapeutic options has some limitations due to the high variability of flu, so vaccines do not always possess all the antigenic characteristics of circulating viral strains and, as far as antiviral treatments are concerned, phenomena of drug resistance may occur. Furthermore, vaccines may have contraindications in patients with a weak immune system, such as the elderly and immunosuppressed individuals, and drugs can lead to some adverse side effects.

Inactivated vaccines are produced by culturing the viral strains in embryonated eggs or eukaryotic cells or using recombinant DNA technologies in which only the HA antigen is expressed in an insect cell line by means of a baculovirus expression system [49,50]. Most inactivated flu vaccines used worldwide are produced by growing Influenza viruses in embryonated eggs, with antigens presented as split or subunits, without adjuvant [51,52].

The flu viruses used in the cell-based vaccines are grown in cultured cells of mammalian origin instead of in hen’s eggs. Cellular technology has the potential for a rapid Influenza vaccine manufacturing process. For recombinant flu vaccines, four different baculoviruses are used for a quadrivalent vaccine that expresses the HA of two IAVs and the HA of two IBVs. The hemagglutinins are then extracted from the infected cells and purified. Currently, only one flu vaccine is manufactured using FDA-approved recombinant technology in the United States. This production process is faster than the previous ones as it is not limited by the selection of vaccine viruses suitable for growth in eggs or for the development of cell-based vaccine viruses.

Concerning the production of the different types of vaccines, standardization is based on HA content, with most vaccines containing 15 μg of each HA antigen. For vaccine production, strains are selected based on predictions made from surveillance data acquired under the coordination of the WHO Global Influenza Surveillance and Response System (GISRS). In general, the majority of available vaccines are quadrivalent, containing an A(H3N2) and an A(H1N1)pdm09 virus strain, and a representative of the two IBV lineages circulating (B/Yamagata and B/Victoria), however, there are also trivalent vaccines, containing the two IAV strains and one IBV lineage. Inactivated, trivalent, or quadrivalent Influenza vaccines primarily induce an antibody response against hemagglutinin and viral neuraminidase. While representing the best tools for preventing Influenza and its complications, these vaccines often fail to induce high protective effectiveness and need annual updates to keep up with evolving new Influenza strains. In fact, the effectiveness of the vaccine decreases when there is a discrepancy between its composition and circulating Influenza viral strains.

Antivirals represent another approach to fighting the different subtypes of Influenza viruses. At present, there are only three classes of chemotherapeutic drugs approved by Food and Drug Administration (FDA) for the treatment of Influenza virus infections: the NA inhibitors oseltamivir phosphate (Tamiflu^®^ and generic), zanamivir (Relenza^®^), peramivir (Rapivab^®^), Laninamivir Octanoate Hydrate (Inavir^®^ and generic); the M2 ion channel blockers amantadine (generic) and rimantadine (Flumadine^®^ and generic); and the cap-dependent endonuclease inhibitor baloxavir marboxil (Xofluza^®^). Oseltamivir phosphate is approved to treat Influenza in adults and children (including full-term newborns). Moreover, to prevent Influenza, it can be used in patients one year of age and older who have been in contact with someone who has flu and when flu is circulating in the community. Zanamivir is approved for the treatment of Influenza in patients seven years and older. It is also approved for the prevention of Influenza in patients five years of age and older. Since this product is inhaled, it is not recommended for people who have respiratory diseases such as chronic obstructive pulmonary disease or asthma. Peramivir is approved to treat acute uncomplicated Influenza in people aged two years and older who are shown to be symptomatic for no more than two days. This drug is given intravenously by a healthcare provider. Baloxavir is approved to treat flu in children aged five years to under twelve years who do not have any chronic medical conditions, and for all people twelve years old and older. This drug also is approved for post-exposure prophylaxis of flu in people aged 5 years and older. Since most anti-Influenza medications target viral proteins, their efficacy may only be strain-specific and may vary in response to future viral mutations [53,54]. Unfortunately, in recent years, Influenza viruses have been frequently developing resistance to licensed drugs. As an example, all IAVs currently in circulation are resistant to the adamantane antivirals amantadine and rimantadine [52], which are, therefore, not recommended for monotherapy. A meta-analysis reported that oseltamivir, zanamivir, peramivir, and baloxavir therapy are associated with a shorter time to alleviate symptoms of uncomplicated Influenza [55], and treatment with oseltamivir or baloxavir significantly reduces antibiotic prescriptions compared with placebo [56]. The emergence of oseltamivir resistance, when it appears, occurs slightly more frequently in Influenza A(H1N1)pdm09 virus infections than A(H3N2) virus infections [57]. The emergence of resistance to oseltamivir is high for severely immunocompromised patients, and who have prolonged Influenza viral replication [58]. Resistance to baloxavir can emerge rapidly, more commonly in viruses A (H3N2) than in virus A (H1N1) pdm09, and is more common in young children [59]. In addition to the resistance phenomena described, authorized drugs can have undesirable or toxic side effects [60]. For example, nausea, vomiting, nosebleeds, headaches, and fatigue are possible adverse effects of oseltamivir phosphate [61,62]; headaches, nauseousness, diarrhea, nose irritation, and vomiting are typical zanamavir side effects and diarrhea is a frequent adverse effect of peramivir. Typical adverse effects of baloxavir are bronchitis, nausea, headaches, and diarrhea [63]. There is no information on baloxavir’s use in patients who are pregnant, nursing mothers, outpatients with severe or progressing illnesses, or hospitalized patients.

In summary, vaccines do not always possess all the antigenic characteristics of circulating viral strains because of the production times, and the vaccine may also have counter-conditions in people with a reduced immune system. On the other hand, the threat of the spread of new flu strains, and increasing resistance to conventional antiviral drugs, have encouraged the search for alternative treatments for IAVs [64,65]. Here, we review the antiviral strategies of peptides as a means to achieve this result.

## 2. Peptides as Drugs

Peptides are therapeutic molecules made up of an amino acid chain, and their molecular weight is between 400 and 5000 Da [66]. The first peptides were born thanks to the study of natural hormones. In fact, in 1921, insulin was developed as the first peptide drug to treat diabetes. Subsequently, with the advent of new synthetic technologies and recombinant DNA techniques, there has been an increase in the design and development of peptides. Despite their selectivity of action and low toxicity, peptides have limitations such as a short half-life, chemical–physical instability, rapid elimination, and low oral bioavailability. To overcome these obstacles, chemical modifications have been made: the use of D amino acids [67]; unnatural amino acids [68]; N-terminal capping; head-to-tail cycling; extension of N terminals or C terminals [69], and peptidomimetics. Moreover, now the pharmaceutical world is currently giving a lot of attention to peptides, with more than 80 marketed peptides and about 500 in pre-clinical trials. Many other hormones have contributed to the birth of peptides, including oxytocin, vasopressin, and somatostatin [70]. Desmopressin, which is a synthetic analogue of vasopressin, was, on the other hand, produced with the new synthetic techniques. While synthetic somatostatin was developed thanks to SAR studies, by the Ala scan technique, each amino acid was replaced with alanine until the pharmacophore was identified [71]. Head-to-tail cyclization allowed the synthesis of Pasireotide, which is more stable than somatostatin. In 2019, two other peptide drugs, gonadotropin-releasing hormone agonists, Leuprolide acetate, and Goserelin, were sold on a large scale, billing pharmaceutical companies over a billion dollars. In addition to hormones, antiviral peptides such as Enfuvirtide, composed of 36 amino acids, structurally similar to the human immunodeficiency virus HIV proteins, have been developed and successfully used for HIV treatment [72]. No less important are the therapeutic peptides of natural origin, including Ziconotide, an analgesic peptide extracted from the marine snail Conus magus, used to treat severe chronic pain [73]. Due to the poor pharmacokinetic profile of the peptides and the poor oral bioavailability, the main routes of administration are the subcutaneous and intramuscular routes. Fortunately, many advances have been made from this point of view, and now there are different administration routes: slow-release subcutaneous injection, intranasal, and, finally, oral Semaglutide and many others [74]. Peptide drugs currently cover a wide therapeutic area: pulmonology, urology, metabolic, cardiovascular, and antimicrobial.

In particular, the COVID-19 pandemic has placed an increasing demand on new antiviral drugs with broad-spectrum activity [75,76].

Antiviral peptides (AVPs) as therapeutics show antiviral effects by directly inhibiting the virus with different mechanisms of action [77]. Unfortunately, the increase of viral resistance [78], concomitant viral infections [79], and viral outbreaks such as COVID-19, H1N1, Ebola virus, and Zika (ZIKV) [80,81,82], highlight the low efficacy of antiviral treatments.

Exploitation of peptides as antivirals is an intense field of research, as demonstrated by recent literature reviews [83,84,85,86,87,88].

In this context, we reviewed literature focusing on antiviral peptides acting on Influenza A virus. These compounds have been classified on the basis of the interacting target in peptides binding HA, NA, PB1, and others exploiting a different mechanism of action.

## 3. Peptides Targeting HA

HA, which belongs to the class I fusion protein family, is one of the proteins expressed on the surface of the viral particle along with NA. Today, eighteen HA subtypes have been identified and subdivided into two phylogenetic groups: group 1 and group 2 (Figure 3A). Structurally, HA is a large mushroom-like homotrimeric protein (Figure 3B) [89]. Each monomer is expressed as a single-chain precursor (HA0) hydrolyzed by different proteases into two-disulfide bound chains (HA1 and HA2), rendering it fusogenic. HA1 constitutes the apical globular head and is responsible for the virus uptake into an endosome. In fact, on the top of the globular head, the receptor binding site (RBS) recognizes the sialic acid (SA) of host membrane glycoproteins. This is a multivalent interaction, as each monomer binds one sialic acid molecule with low affinity, but the contemporary binding of more SA increases the affinity and stability of binding [90].

The HA2 chain constitutes the stem portion of this mushroom-like protein. It is very conserved among HAs because it contains the fusion peptide (1–15 aa). Endosomal acidification causes the conformational changes of HA2 to form the pre-hairpin structure, exposing the hydrophobic fusion peptide. Low pH induces a conformational transition (loop-to-helix) of the inter-helical loop (B loop), allowing the central coiled-coil to be extended and the fusion peptide to be relocated. The N-terminal HA2 fusion peptide (FP) can thus be inserted into the host membrane. Following protein refolding, the two membranes (host and viral) were brought together to fuse [91].

RBS and fusion peptide are key players in the HA-mediated processes and could be viable targets for anti-IAV drug development. Many peptides have been discovered to block viral entry to the host cell. They can be classified based on the binding region of HA and, therefore, the hindered process mediated by HA. There can be compounds competing with the sialic acid binding at the RBS, or compounds that interact with other regions of HA, hampering the conformational rearrangement of the protein and impeding the internalization process. Other peptides can block viral entry using different mechanisms not involving the HA binding and are reported at the end of this paragraph.

### 3.1. Peptides Interfering with the Sialic Acid Binding

Jones and coworkers explored a library of 5 cell-penetrating peptides and identified a 20 aa Entry Blocker (EB) peptide obtained from the fibroblast growth factor 4 (FGF-4) signal sequence. The peptide antiviral activity was assessed on Madin–Darby canine kidney (MDCK) cells infected with A/Hong Kong/483/97 (H5N1) viral strain and in vivo on Balb/c mice infected with the same virus. The peptide was demonstrated to interfere with the viral binding to the host cell through hemagglutination inhibition (HI) assay and showed activity in the micromolar range toward several IAV strains (H1N1, H2N2, H3N2, H5N1, H5N9, H7N3). The low therapeutic index of 22 limited the exploitation of this peptide [92]. The same authors identified in a subsequent studied the minimum peptide sequence, maintaining the antiviral activity of the lead EB constituted by 13 aa (B10) and a newly designed peptide of 16 aa (B7), endowed with a micromolar activity on MDCK cells infected with PR/8 (H1N1) virus and an increased selectivity index [93]. Further studies demonstrated that the EB peptide expresses its activity by causing viral particle aggregation, and therefore it has also been proposed as a vaccine adjuvant, able to boost phagocytosis by macrophages [94]. To complete the picture, Lu et al. studied the binding of EB peptide to H1 and H5 HAs by docking and mass spectrometry. Blind docking calculations were carried out on the head region of HA to account for the HI activity shown by this compound. The authors hypothesized more than one possible binding mode for each HA close to the receptor binding site, justifying HI activity. Moreover, they studied the EB-peptide:HA binding by mass spectrometry, evaluating the stabilization effect that EB peptide binding produces on HA hydrolysis. Experimentally obtained data superimposed with docking results suggested contact at the 220-loop region of hemagglutinin [95]. Recently, Reyes-Barrera et al. introduced nucleotide sequence of the EB peptide into the nuclear genome of microalgae Chlamydomonas reinhardtii to reduce the peptide production costs. They observed that the EB peptide extract from the microalgae was 100-fold more effective than the EB synthetic peptide to prevent HA activity of Influenza A/H1N1 pdm and Influenza A/Virginia/ATCC/2009 (H1N1) strains. Additionally, they examined the ability of these peptides to affect the virus replication in MDCK cells by neutralization assay, and the EB peptide extract had a 32-fold greater antiviral potency than the synthetic peptide against Influenza A/H1N1 pdm (IC50 values: 20.7 nM and 754.4 nM, respectively). It must be noticed that the EB peptide sequence expressed in microalgae is ten amino acids longer, which can contribute to binding or peptide stability. Moreover, the expressed EB peptide is more soluble than synthetic, improving its bioavailability [96].

Along with EB, peptides mimicking sialic acid obtained by phage display screening by Matsubara and coworkers represent reference peptides binding HA. A random library of 15-mer peptides was preliminarily explored by affinity selection with both H1 and H3 HAs, belonging to group 1 and group 2 of the phylogenetic tree, to find broad-spectrum compounds. After a second selection, the authors identified and resynthesized four peptides, and their binding to HA was assessed by surface plasmon resonance (SPR) analysis. The most active s2 peptide was submitted to Ala-scan and fragmentation, obtaining 5-mer peptides that increased the protective effect against IAV infection (ARLPR). The overall identified eight active peptides were conjugated to stearic acid to favor the antiviral activity. N-stearoyl peptides, in fact, should assemble in supramolecular systems, such as micelles, that are expected to improve their activity [97]. C18-peptides showed low micromolar activity toward MDCK cells infected with A/Puerto Rico/8/34 (H1N1) and A/Aichi/2/68 (H3N2) viral strains in plaque reduction assays. Docking experiments elucidated the binding mode of the smaller fragment into the RBS of H3 HA, where the ligand forms four H-bonds and establishes hydrophobic contact with surrounding residues [98]. Peptides identified by Matsubara were exploited in a series of following papers, such as the one from Huttl et al. that synthesized three peptides derived from the previous article (L1, ARLPRTMVHPKPAQP; M1, ARLPRTMV; S1, ARLPR) and their palmitoyl derivatives to obtain peptide amphiphiles, able to form supramolecular systems. The formation of micelles or other larger molecular assemblies is often used to improve the binding with HA because it should allow the contemporary binding to the three sialic acid binding sites. In this work, the authors verified the improved affinity to HA of palmitoylated peptides with respect to simple peptides. Compound binding was assayed via SPR, measuring the signal obtained from the interaction with HA from the H5N1 strain immobilized on a sensor. Palmitoylated L1 peptide showed a tenfold increased signal compared to the simple peptide at 500 µM. No antiviral activity was evaluated [99]. Following the same rationale of multimeric binding to HA, the identified peptide mimicking sialic acid (ARLPR) was further used to functionalize carbosilane-based dendrimers with different frameworks (fan, ball, dumbbell). Dendrimers are also meant to occupy the three binding sites present on the trimeric structure of HA, exploiting the “clustering effect”. Moreover, they are flexible, chemically stable, and biologically acceptable. Fan and dumbbell-shaped dendrimers containing three and six peptide units, respectively, exhibited protective activity toward MDCK cells infected with a mixture of A/Puerto Rico/8/34 (H1N1) and A/Aichi/2/68 (H3N2). Dumbbell-shaped dendrimers were the most potent with a subnanomolar IC50, while both structures presented negligible toxicity [100]. In another article, the ability of the pentapeptide ARLPR to mimic sialic acid was exploited to build a system for HA and Influenza virus capture and detection. To this aim, the peptide was conjugated to DPPE to form a lipid bilayer, then subsequently immobilized on mica or plastic plates. This system was effective in binding H1 HA and Influenza virus similarly to commonly used glycoconjugates [101]. The same authors repeated the previous protocol to identify heptapeptides mimicking sialic acid. A random phage display library of 7-mer peptides was screened by affinity selection, obtaining the peptide LVRPLAL that showed the highest affinity toward H1 and H3 HAs. The avidin–biotin-peroxidase complex (ABC) system was exploited to evaluate the binding of this peptide to HA. It showed a Kd of 92 and 194 µM for H1 HA from A/New Caledonia/20/99 (H1N1) and H3 from A/Wyoming/3/2003 (H3N2), respectively. The C18-conjugated form of this peptide was tested in plaque reduction assay, showing a good activity toward A/Aichi/2/68 (H3N2) virus (6.4 µM) and a lower potency against A/Puerto Rico/8/34 (H1N1) virus (101 µM). The Ala scan indicated the importance of Arg3 and Pro4 for the HA binding, which was also assessed by docking calculations [102].

Utilizing other species’ host innate defense mechanisms is a valuable method for developing antimicrobials [103,104]. All species have a vital natural defense mechanism known as host defense peptides (HDPs), which is much less susceptible to resistance than traditional pharmacological treatments. In this field, a high-mannose lectin was extracted from the green alga Boodlea coacta (BCA), sequenced, and studied as an anti-HIV and anti-Influenza agent. This 180 aa glycoprotein showed nanomolar efficacy in blocking infection of 9 strains of H1N1 and H3N2 IAV and one IBV on MDCK cells. This small protein was demonstrated to interact with HA through HI assays, but this contact can be attributed to sugars monomers rather than protein interaction [105].

In 2011, Li et al. explored the potential antiviral activity of mucroporin, a cation peptide derived from the scorpion venom, and its optimized version mucroporin-M1. The latter, but not original mucroporin, showed activity toward Measles virus, SARS-CoV, and H5N1 IAV at a micromolar level. As the peptide acts in the first 40 min of infection, it is plausible to act during the first entry phases [106].

Innate immunity proteins inspired a series of studies on the antiviral activity of lactoferrin (Lf) and its derived peptides. Lactoferrin is secreted in mammalian milk, saliva, and tears, and contributes to innate immunity thanks to its solid anti-infective activity on bacteria, fungi, and viruses [107]. Pietrantoni et al. demonstrated the anti-Influenza activity of bovine lactoferrin (bLf) that protects MDCK cells from apoptosis induced by the virus [108]. Moreover, it has been demonstrated that bLf iron saturation and glycosylation do not affect the anti-Influenza action of this protein [109]. Prompted by these results, the study focused on the single lobes composing bLf. However, while N-lobe was ineffective, the C-lobe retained the bLf activity on different viral strains (H1N1, H3N2, H5N1, H7N1) from femto to nanomolar concentration in hemagglutination inhibition (HI) assays, and picomolar on MDCK cells infected with H1N1 and H3N2 strains. A Western-blot run followed by sequencing indicated the interaction between bLf C-lobe and the fusion peptide of HA. Protein–protein docking calculations were carried out to mimic the contact between HA and bLf C-lobe and indicated the possible role of three bLf C-lobe loops in the binding to viral hemagglutinin. The three peptides (SKHSSLDCVLRP (P1, aa 418–429), NGESTADWAKN (P2, aa 552–563), and AGDDQGLDKCVPNSKEK (P3, aa 506–522)) were tested and confirmed their efficacy on the same viral strains mentioned previously, but with increased activity ranging from femto to picomolar in HI tests, and cell-based assays. Moreover, they presented no toxicity until the concentration of 25 ∝ M, with a selectivity index of 106–108 [110,111]. The activity expressed by these peptides on H1N1 and H3N2 strains accounts for the broad-spectrum activity of these compounds that block phylogenetically far HAs. Furthermore, they are, at least to our knowledge, the most potent peptides in cell-based assays. The P1 peptide was further investigated by Scala et al., which synthesized a series of elongated and reduced peptides starting from the sequence SKHSSLDCVLRP. Moreover, other peptides corresponding to other accessible loops of bLf C-lobe were synthesized to screen for their possible antiviral activity. Eight further peptides were identified and endowed with picomolar antiviral activity on MDCK cells infected with two H1N1 and one H3N2 strain, confirming their broad-spectrum activity. SPR analysis further assessed the high-affinity binding of studied peptides to HA, while NMR structural analysis was carried out to study their conformational preferences [112]. Moving from the identified potent tetrapeptides SLDC and SKHS, an Ala scan approach led to the synthesis and biological evaluation of further eight peptides, studied by SPR and microscale thermophoresis, HI, and cell-based assays. The tetrapeptide SAHS showed the most promising profile with a subnanomolar broad-spectrum antiviral activity. Docking studies suggested the binding of studied peptides on the RBS of HAs, where they compete with sialic acid, as demonstrated by the HI activity [113].

Memczak and coworkers identified three peptides derived from mAb binding HA of Aichi H3N2. The analysis of the PDB structure 2VIR highlighted the interacting role of three mAb loops binding the sialic acid binding site (PaA, SGFLLISNGVHWV; PeB, ARDFYDYDVFYYAMD; PeC, LGVIWAGGNTNY). Before the experimental testing, the putative binding of the three peptides with HA was assessed by MD simulation. Computationally calculated binding ΔG suggested the promising binding of PeB. SPR binding assay on the three peptides confirmed the MD prediction, with PeB being the most potent, followed by PeC, while PeA was inactive. PeB peptide was optimized by site-directed substitution: 152 new variants were generated, and binding to more H1N1 and H3N2 strains was assessed. The most interesting peptide was the PeBGF, which showed a micromolar activity in SPR, HI, neutralization, and infection inhibition assays toward both Aichi H3N2 and Rostock H7N1 viral strains [114]. In the following article, the researchers explored the effect of conjugation of one of the previously developed peptides PeBGF with stearic acid, analogously to what was done by Matsubara et al. [98]. The acylated peptide was able to block hemagglutination produced by the Aichi H3N2 and Rostock H7N1 viruses at 1.2 µM and 2.8 µM, respectively, increasing the activity with respect to the non-acylated peptide by ten folds. MDCK cells’ infection inhibition was in the same order of magnitude. However, the authors observed the ability of these structures to form supramolecular systems, such as fibers and sheets and not micelles, and to cause red blood cell agglutination by themselves because of their attitude to interact with membranes [115]. To overcome this limitation, the same authors covalently conjugated the peptide ligand to polyglycerol (PG)-based hydrophilic dendritic scaffolds with different molecular weights and degrees of functionalization for peptide conjugation. Obtained nanoparticles increased the antiviral activity compared to simple peptides of almost three orders of magnitude passing from micromolar to nanomolar IC50 in both HI and infection inhibition with X31. In vivo tests on Balb/c mice infected with X31 demonstrated the efficacy of the two most active constructs 4b and 4d, with increased protecting activity with respect to siallyllactose presenting PAMAM dendrimers [116].

Phage display screening is among the most effective method to identify active peptides binding to a protein epitope. A heptapeptide phage display library was submitted to affinity selection against H9N2 avian virus particles. This procedure allowed the identification of the peptide NDFRSKT that targets HA and, along with its cyclic form (CNDFRSKTC), showed antiviral activity at high micromolar concentration and no cytotoxicity [117]. The authors assessed the mechanism of action of this peptide, verifying that it interferes with the host cell attachment and not with the fusion process [118].

In a recent study, Arbi and coworkers exposed the A/chicken/Tunisia/12/2010 (H9N2) avian virus to a phage display library of linear hexapeptides in three rounds. Sixteen selected peptides were tested in HI assays, and the antiviral efficacy of thirteen active peptides was assessed in vivo. Two peptides (P1, and P2) out of thirteen showed antiviral activity and were administered in vivo to chickens. Peptide P1 presented a protective effect on 80% of treated chickens until five days post-inoculation. Moreover, the two peptides did not show toxicity on eggs and chicken lungs. Molecular docking calculations, carried out to depict the interaction between active peptides and hemagglutinin, indicated that P1 and P2 peptides bind different regions of the HA globular head: P1 occupies the RBS, while P2 occupies a site close to the RBS and interacts with the 220-loop [119].

Interesting work from a Japanese research group reported a screening approach exploring a library of macrocyclic peptides active toward avian IAV. Cyclic peptides were selected using an innovative screening approach named RaPID system, which integrates mRNA display technology with the Flexible In vitro Translation (FIT) system. Two thioether cyclic peptide libraries were built using proteinogenic amino acids or 11 proteinogenic and four N-methyl-amino acids to improve the stability of obtained peptides. Iterative rounds were carried out on recombinant HA from avian Influenza virus A/Vietnam/1203/04 (H5N1) or A/Bar-Headed goose/Qinghai Lake/1 A/05 (H5N1). Twenty-eight resulting binders were submitted to plaque reduction assay, and two macrocycles (iHA-24 and iHA-100) turned out to be the most active against three avian H5N1, but also against H1N1 and H2N2 strains. Further studies assessed the ability of iHA-100 to block viral adsorption and fusion processes. Escape mutant isolation demonstrated its binding at the stalk region of HA, confirmed by the stabilization effect its binding produces on HA trypsin degradation, but interaction on the globular head was not excluded. The macrocycle peptide was tested in vivo on mice infected with lethal H5N1 IAV. iHA-100 rescued 40% of mice and showed better efficacy when administered in the early or late phases of infection. In vivo testing also involved non-human primate cynomolgus macaque infected with H5N1/Vietnam strain and temperature monitored. iHA-100 administration reduced the primate temperature rise and bodyweight loss [120].

In 2020, Omi et al. explored a tetravalent peptide library characterized by a polylysine scaffold, a spacer, and four randomized peptides connected to the core. They identified a peptide able to bind viral HA exploiting the “clustering effect”. All tetravalent peptides could interfere with the recognition between HA and α2,3-sialyllactose polymer, a mimic of the HA receptor, suggesting their direct binding to the RBS of HA. The same peptides were assayed in cytopathic effect inhibition, and PVF-tet (RRPVNHF) presented the highest activity. Surprisingly, this compound did not block viral entry into the cell as expected because of its binding to HA and did not act on the fusion process or in the HI assay. In fact, PVF-tet does not interact with H1 and H2, but likely does with the H0 of newly synthesized viral particles. PVF-tet is a cell-penetrating peptide, and its addition to the infected cell causes the amassment of HA in a vacuole-like structures named amphisomes [121]. A synthetic report of the most representative peptides binding the head of HA is presented in Table 2.

### 3.2. Peptides Interfering with the Fusogenic Activity of HA

In 2005, Zhao et al. exploited affinity chromatography to identify antisense peptides able to interact with the fusion peptide (FP) of HA. The explored library was derived from a previously identified sequence (YRSKQA) [122]. The FP was immobilized on the column, and a synthesized library of 108 peptides was tested, determining a preferred peptide binding FP. Starting from this sequence, more elongated peptides were synthesized and assessed in affinity chromatography experiments, indicating the possible antiviral activity of the hendecapeptide FHRKKGRGKHK, tested on MDCK cells infected with two H1N1 strains [123].

Wu et al. also focused on the FP, specifically the last 23 amino acids of the N-terminal portion of HA2. The comparison of the FPs of the 16 IAV strains revealed their hydrophobicity, the content of multiple glycine residues, an overall negative charge, and the highly conserved residues 1–11. Moreover, the secondary structure is an amphipathic α-helix or partial α-helix. To find antiviral peptides (AVPs), Wu et al. generated pseudo-fusion peptides (pFPs) that could interact with the HA2 subunit by substituting lysine with negative or neutral residues, generating “positively charged” pFPs. The antiviral activity of the pFPs was assessed against IAV strains Puerto Rico/8/34 (H1N1) and Aichi/2/68 (H3N2), and the mechanism was investigated using a variety of pseudovirus-based assays. As revealed by fusion and hemolysis inhibition assays, these peptides specifically blocked the entry of IAV into host cells via the interaction of the pFPs and the HA2 subunit, most likely due to interactions between the N-terminal portion of HA2. The positively charged lysine residues of pFPs strongly interact with glutamic acid or aspartic acid, negatively charged, of the HA2 subunit via ionic contacts and hydrogen bonds [124].

Lopez-Martinez et al. focused on the stalk region formed by HA2 and its FP and the N- and C-terminal residues of HA1 (the F’ fusion subdomain). By in silico analysis of different HA1 and HA2, they discovered conserved regions (40–50 amino acids at the N- and C-terminal of HA1, and 80 amino acids at the N-terminal of HA2) that revealed short regions of HA1 N- and C-terminal ends characterized by hydrophilic, flexible, charged, and exposed (antigenic) sequences. Inhibitory activity of those peptides against IAV of various origins (human, swine, and avian) demonstrates the ability to inhibit IAV without cytotoxicity. The binding to the HA, which prevents the conformational rearrangements inside the endosome, is the most likely mechanism of action. Docking studies revealed that AVPs form several hydrogen bonds and electrostatic contacts in the HA stalk region with the fusion peptide, helix A, helix B, and loop B that could be responsible for the inhibition of HA conformational changes. Nevertheless, they did not show the stability at endosomal pH [125].

Lee et al. targeted the post-fusion structure of the HA2, generating three peptides equivalent to residues 155–185 (P155–185), 155–181 (P155–181), and 155–175 (P155–175) of the HA2 sequence, and their respective cholesterol-tagged by cysteine mediated link (P155–185-Chol, P155–181-Chol, P155–175-Chol). The complete sequence (155–185) corresponds to the HA2 portion that bundles to the inner coiled-coil that is crucial for the fusion process; compound P155–181 lacks the four final residues between the N-capped coiled-coil and the transmembrane domain, while compound P155–175 corresponds to the most truncated sequence deficient the N-cap motif. Only cholesterol-tagged compounds containing the N-cap motif (155–185-Chol and 155–181-Chol) demonstrated antiviral activity against IAV/H3N2 in cell culture, most likely due to the ability of the cholesterol to fasten the peptide to the membrane and to form nanoparticles [126]. In a subsequent article, the same authors created a 43-amino acid fusion inhibitory peptide (HA2Ec1) coming from the α-helical region of the HA2 that participates to helix-helix contacts. In particular, the peptide identified by Lee as P155–185 was expanded by adding 12 residues from the N-terminus, yielding a new peptide corresponding to the sequence 143–185. This peptide was also conjugated with a flexible polyethylene glycol (PEG) linker to cholesterol (Chol) or tocopherol (Toc) (HA2Ec2 and HA2Ec3) to improve its concentration on the membrane before virus endocytosis. Three more peptides (Tat-HA2Ec) contained a cell-penetrating peptide obtained from the HIV-1 Tat protein to improve its internalization. The addition of Tat sequence increased the inhibition of Influenza fusion with liposomes triggered by pH, while the Chol and Toc conjugation favored the kinetics. The peptides were non-cytotoxic in an ex vivo model and may be safe for in vivo use. In solution, the compounds self-assemble to form small peptide nanoparticles (NPs) that are suitable for efficient in vivo biodistribution. The in vivo efficacy of the NPs was comparable to that of the widely used neuraminidase inhibitor zanamivir. Even though the local viral titer in infected rats decreased significantly, intranasal administration could be an option for fusion peptide NP delivery. The NPs produced viable fusion inhibitors that can be used alone or combined with compounds with different mechanisms of action to control Influenza epidemics [127].

During the fusion process, a leucine zipper-like α-helical hexamer (6HB) formed. Three N-terminal heptad repeat (NHR) regions (adjacent to FP) formed a central trimeric helix scaffold (the pre-hairpin conformation) during the coiled-coil 6HB assembly, then three C-terminal heptad repeat (CHR) regions (immediately precedes the transmembrane domain) wrapped in antiparallel orientation onto the periphery of the core. The inhibition of the coiled coil-mediated interactions NHR/CHR is a promising approach for the development of broad-spectrum agents. Bioactive peptides derived from the CHR region act as decoy α-helices, forming a heterologous non-functional 6HB structure and inhibiting virus–host cell membrane fusion.

Wang et al. replicated the topography of CHR helices by designing amphiphilic α-helical peptides conjugated with fatty acids. The amphiphilic peptides are based on a sequence of five heptads, followed by β-Ala, and Lys bounded to a palmitoyl group (C16), useful for membrane-anchoring properties and good safety profile.

The schematic structure of the peptides [Ac-(XaEbEcXdZeKfKg)5-βAla-K(C16)NH2] contains hydrophobic residues (X) and polar/charged residues (Z) at the foreground positions (a,d,e) and a combination of glutamate (E) and lysine (K) residues at background positions (b,c,f,g) required to form double E-K intra-strand ionic contact at i to i + 4 positions to promote the overall α-helicity and solubility of the peptides. IAV Puerto Rico/8/34 and Hong Kong/8/68 were inhibited by the lipopeptide IIQ (EC50 1.73 μM and 0.70, respectively). All of these compounds were not cytotoxic on MDCK cells. The viral replicon and neuraminidase inhibition assays revealed no effect. The hemagglutination assay established that the binding target is not on the HA1 subunit. In contrast, the hemolysis inhibition assay established that IIQ can interact with the HA2 subunit, disrupting conformational changes or interaction with the HA1 subunit by inhibiting virus absorption into host cells. Moreover, the interaction of IIQ with a synthetic peptide containing the NHR segment or the post-fusion structure of the H3N2 HA2 subunit revealed that compound IIQ associates with a site in the NHR region, generating a heterogeneous 6HB structure that blocks the fusion process [128].

Koday and colleagues designed a novel 97-residues protein (HB36.6) that binds the highly conserved HA stem with high affinity, based on a previously computationally designed HA stem-binding protein HB36.5 which mimics the broadly neutralizing monoclonal antibodies (bnAbs) FI6v3 [129]. Compound HB36.6 neutralizes various group 1 H1 and H5 viruses in vitro, whereas in vivo, intranasal administration protects against three extremely different H1 and H5 Influenza strains. Furthermore, HB36.6 was discovered to intervene in the Influenza protection independent of the host response, and the pre-exposure treatment avoided infection without an inflammatory response, lowering the risk of disease exacerbation due to immune effector-mediated inflammation. The possibility of use in immune-compromised or elderly patients should be highlighted. This compound emerged as the starting point for a new class of antivirals targeting the HA-stem for prophylactic and therapeutic use [130].

Kadam et al. concentrated on developing a small peptide that could prevent HA conformational changes at low endosomal pH. The peptides were generated using complementarity-determining region (CDR) loops from human antibodies to HA FI6v3 and CR9114, which binds to a hydrophobic region at the interface of HA1 and HA2 in the HA stem that is highly conserved. A series of linear synthesized peptides were tested for binding to a panel of HAs from various IAV strains. Peptide 1 (P1), identical to heavy chain 3 of FI6v3 except for a Glu4 instead of a Leu, was identified as the lead. P1 was used to create a library of cyclic lactam peptides containing non-proteinogenic amino acids. Peptide P7 had a similar affinity, potency, and virus neutralization of H1N1 and H5N1 viruses as other peptides, but the complex P7-HA was more stable. The crystal structures of the complexes proved the effective modes of binding of the peptides to the HA stem even at low pH, confirming the ability of these new peptide-based small molecules to inhibit the conformational changes of the HA trimer in the endosome and to prevent the trafficking to late endosomes [131].

In 2012, Nicol et al. disclosed the antiviral activity of a series of peptides named FluPep (FP). The lead peptide, Tkip (FP1), is a 12-mer mimetic of the suppressor of cytokine signaling (SOCS) protein. Six further peptides were synthesized to improve the lead compound solubility and stability or assess the truncated form activity. All seven compounds were tested in plaque reduction assay on MDCK cells infected with A/WSN/33 (H1N1) strains. All peptides were actives, with FP3 and FP4 presenting the highest potency (0.03 and 0.04 nM, respectively). Notably, most active peptides FP3 and FP4 share the RRKK sequence characterizing the EB peptides. The mechanism of action was demonstrated to be the interference with the viral attachment via contact with HA, as demonstrated by ELISA assay, but they do not block hemagglutination produced by the virus, so they do not interact on the receptor binding site. Moreover, the compounds were tested on Balb/c mice infected with a lethal dose of A/WSN/33 virus, showing an ability to reduce the viral titer by 2–4 log units [132].

An example of a host defense peptide is represented by urumin, a 27-mer cationic peptide, secreted by the South Indian frog Hydrophylax bahuvistara on its skin, which binds the stalk region of HA and is active toward the H1 subtype. Thirty-two peptides secreted by frogs were screened through plaque reduction assay on A/Puerto Rico/8/1934 (H1N1) strain. Four peptides showed activity; among them, urumin showed the best activity (3.6 µM) and the lowest toxicity (20% at 1.430 µM on red blood cells). Further tests verified its activity on other H1N1 strains and H3N2 IAV. Urumin was confirmed to be active toward H1N1 viruses and less potent on the phylogenetically far H3N2 strain. To validate its target, urumin was tested against four A/Puerto Rico/8/1934 (PR8) reassortant viruses (H1N1, H1N2, H3N1, and H3N2), conserving its efficacy only toward the H1 containing strains and indicating the HA and the viral target macromolecule. Moreover, the action on chimeric H9N3 virus, with the HA head region from the H9 strain and the stalk from PR8 H1, confirmed its binding on the stem conserved region of HA [133].

The strategy to conjugate peptides with a lipid chain was also applied by Wu et al., which focused on identifying IAV entry inhibitors. Two peptides, C12-KKWK and C12-OOWO, were submitted to several assays to verify their anti-Influenza action. Both compounds showed antiviral activity, with IC50 values of C12-KKWK and C12-OOWO of 7.30 ± 1.57 and 8.48 ± 0.74 mg/L against PR8 (H1N1), and 6.14 ± 1.45 and 7.22 ± 0.67 mg/L against A/Aichi/2/68 (H3N2), respectively. Moreover, they have been shown to be unable to inhibit neuraminidase but have presented anti-hemolytic activity. Hemolysis is caused by the conformational rearrangement of HA triggered by acidic pH to produce membrane fusion. Therefore, this activity accounts for interaction with the conserved stem region of HA, as confirmed by the inactivity in HI assay [134]. In the following article, the authors synthesized a hybridized peptide connecting the KKWK (Hp) sequence with the ARLPR (Jp) peptide identified by Matsubara et al. [98]. The two fragments were linked in a different order, inserting the GGG sequence and conjugating the resulting peptides with C16-C20 lipid chain. The C20-Jp-Hp (C20-ARLPRKKWK) compound showed the best activity profile and broad-spectrum activity toward both H3N2 and H1N1 IAV strains at a micromolar level in MDCK cell-based assay. Its activity profile was further investigated, assessing that it works on H5N1 pseudovirus. The possible mechanism of action was verified using the same panel of tests described in the previous article. Additionally, in this case, even if the ARLPR is a sialic acid mimic, the compound was demonstrated to bind the fusogenic region of HA, as indicated by CD spectra registered for the fusion peptide alone or in presence of C20-Jp-Hp [135]. In Table 3, most relevant peptides acting on the fusogenic activity of HA are reported.

### 3.3. Entry Blockers Not Interacting with HA

Chen et al. disclosed a series of 18-mer peptides derived from the H1 of HA (HA-pep25) containing the conserved 220-loop of RBD. It showed broad-spectrum antiviral activity toward H1N1, H5N1, and H7N9 pseudoviruses. This peptide was demonstrated to block the host receptor, particularly the oligosaccharide part of sialyl-oligosaccharides on the host cell surface. Shorter and Ala-scan peptides were synthesized and assayed, indicating that the length of the peptide must be maintained, and also indicating the crucial role of eighth amino acids (S1, K2, R9, F12, W14, T15, I16, and K18) for activity [136].

A series of articles explored the antiviral activity of defensins, with small endogenous cationic peptides with antimicrobial activity being classified into three subfamilies: alpha, beta, and theta. Their antimicrobial activity is generally ascribed to the ability to destabilize the membranes, as assessed by Leikina and coworkers. Their work reports the anti-Influenza activity of θ-defensin retrocyclin 2 (RC2), a cyclic octadecapeptide with three disulfide bridges (RRICRCICGRGICRCICG). They demonstrated that RC2 showed antiviral activity on MDCK cells. Further studies indicated that this peptide acts on the entry phases of viral infection, blocking the membrane fusion mediated by the viral hemagglutinin. However, it does not bind HA but interacts with surface glycoproteins, producing crosslinks and hampering membrane fusion [137]. In their study, Salvatore et al. verified the protective role of human α-defensin on MDCK cells infected with A/WSN/33 (H1N1) and A/Panama/2007/99 (H3N2) strains. However, they claimed that it was not due to a cellular mechanism interfering with the post-internalization process, but rather to the interaction with the virus [138]. A subsequent article enlarged the panel of defensins, showing antiviral activity against A/Philippines 82 (H3N2) and PR8 (H1N1) strains. The analysis involved Human Neutrophil peptides (HNP-1, HNP-2, HNP-3, HNP-4), retrocyclins (RC-2, RC-4, RC-101), Human-β-defensins (HBD-1, HBD-2), and Human defensins (HD-5, HD-6). Retrocyclins and HNPs provided the best anti-Influenza activity. The mechanism of action is not due to direct contact with HA but to the ability to produce viral aggregates and increase the IAV intake from neutrophils and macrophages [139]. Different results were presented by Jiang and coworkers, which tested the mouse β-defensin 3 activity against PR8 (H1N1) in vivo. Mice were inoculated with β-defensin plasmid and challenged with intranasal administration of the virus. The bronchoalveolar fluid was titred, and survival was evaluated. Treated mice presented a substantial reduction of lung virus titer and a mice survival rate of 70% [140]. In a successive work, the same authors investigated the activity of mouse β-defensin 2, repeating the same series of biological assays, and demonstrating the protecting activity of this isoform against PR8 H1N1 strain. Treating mice at different times of infection showed that pretreatment with β-defensin 2 was the most effective, indicating its activity in the first phase of infection and blocking viral entry in the host cell [141]. The demonstrated antiviral activity of mouse- β-defensin 4 (mBD-4) prompted Zhao et al. to synthesize a series of reduced peptides that allow identifying a 30-mer peptide (P9) with increased antiviral activity and lower toxicity with respect to staring peptide. P9 showed a broad-spectrum antiviral activity on several IAV viral strains (H1N1, H3N2, H5N1, H7N7, and H7N9), and on two coronaviruses, SARS-CoV and MERS-CoV. This peptide totally rescued H1N1 lethally challenged mice. Concerning the mechanism of action, it showed its activity in the early phases of infection; therefore, it is supposed to bind surface glycoproteins such as HA. Thanks to its content of basic amino acids, P9 prevents endosome acidification, blocking the HA conformational rearrangement and the fusion process [142]. The same mechanism of action was recognized for a modified P9 peptide (P9R), with a higher positive charge. P9R showed the same ability as P9 to block endosome acidification and interacted with viral particles of both IAV (H1N1 pdm) and SARS-CoV-2. Because of its mechanism of action, this peptide is active toward different respiratory viruses sharing the mechanism of cell entry. To complete the biological activity profile, P9R was tested in vivo on Balb/c mice infected with A/H1N1/pdm09 and obtained the survival of the 70% of treated animals [143].

Belonging to the same family of endogenous antimicrobial peptides, cathelicidins play a similar role when overexpressed after infection or inflammation. Barlow et al. studied the effect of both human and murine cathelicidin (LL-37 and mCRAMP, respectively) on IAV. Both peptides rescued 70–80% Balb/c mice from lethal infection of PR/8 H1N1 IAV and reduced the weight loss and viral titer in the lungs. LL-37 was demonstrated to act as a virucidal agent, as verified in plaque reduction assay, but it likely also has some immunomodulatory activity [144]. In a subsequent article, Tripathi et al. better examined the LL-37 mechanism of action. This peptide presented antiviral activity on A/Philippines 82 (H3N2) infected MDCK cells via direct binding to the virus, but not viral hemagglutinin. Moreover, this peptide does not bind surfactant protein-D (SP-D), does not increase viral uptake by neutrophils and macrophages, and does not cause virus aggregation, as shown by retrocyclins and HNP [145].

## 4. Peptides Targeting NA

NA is an antigenic glycoprotein anchored in the surface envelope of the Influenza virions and plays a crucial role in virus replication. Thus, it can be considered an excellent drug target for controlling the Influenza infections. Several NA inhibitors are available, such as the inhalant zanamivir and the orally administered oseltamivir [146,147]. They considerably affect the viral infection, but drug resistance remains one of the main problems to be solved [148]. Furthermore, neuraminidase inhibitors trigger significant side effects such as neurotoxicity, gastrointestinal, and respiratory diseases [149]. Thus, new-generation neuraminidase inhibitors are currently under investigation. In particular, great attention is being paid to peptides. Previously, Amri et al. identified cyclic peptides as H1N1 NA inhibitors [150], and Upadhyay et al. showed that the mimosine tetrapeptide (M-FFY) had high NA-inhibitory activity [151]. In 2018, Li and colleagues discovered a novel natural peptide, PGEKGPSGEAGTAGPPGTPGPQGL (peptide P), obtained from cod skin hydrolysates. This compound showed a NA inhibitory activity with Ki of 0.29 mM, and it was demonstrated to directly bind to free enzymes. The cytopathic effect reduction assay showed that the peptide P protected MDCK cells from viral infection and reduced viral production in a dose-dependent manner with an EC50 value of 471 ± 12 μg/mL against PR8 (H1N1) (Table 4). Moreover, time-course analysis showed that it inhibited Influenza virus in the early stage of the infectious cycle. Unfortunately, it was observed that the activity of the peptide was inactivated during the simulated in vitro gastrointestinal digestion, suggesting that oral administration was not recommended [152]. Recently, Chen et al. identified an octapeptide, errKPAQP (P2), from a small library of 20 peptides, designed from the binding pocket of oseltamivir in neuraminidase. P2 showed nanomolar affinity (11 nM) to Influenza neuraminidase and inhibited neuraminidase activity at a concentration of 4.25 μM, leading to effective protection of MDCK cells from Influenza virus-induced death and viral infection. Moreover, errKPAQP presented several advantages: low hemolytic activity, low cytotoxicity, and a good pharmacokinetic profile. Significantly, errKPAQP reduced Influenza virus-induced inflammation and mortality rates in infected mice, indicating that it could protect against the lethal challenge of Influenza viruses in vivo [153].

An interesting study was conducted by Albrecht et al. which evaluated new tools to selectively block plasma membrane NEU1 (mNEU1) sialidase activity by disrupting its dimerization. NEU1 is one of the four types of mammalians sialidases identified and it is characterized by two regions, 139–159 (TM1) and 316–333 (TM2), as potential transmembrane domain. mNEU1 activity is potentially due to its dimerization, so the authors designed interfering peptides (IntPep, ELVDPVVAAGAVVTSSGIVFFS) that target the dimerization interface located in the TM2 domain of mNEU1 and evaluated their effects on its sialidase activity [154]. They used two strategies to deliver the IntPep into cells: TAT sequence (GRKKRRQRRRPQ) conjugation and incorporation of the peptide into LDS micelles [155]. Authors demonstrated, through molecular dynamics simulations and heteronuclear nuclear magnetic resonance (NMR) studies, that these IntPep peptides were able to interact with the TM2 domain (region 316–333) of human NEU1 and disrupt its dimerization with consequent losses in sialidase activity. Furthermore, these IntPeps inhibit membrane sialidase activity triggered by elastin-derived peptides in macrophages, which are highly dependent on NEU1 [156].

**Table 4 ijms-23-11433-t004:** Most representative peptides targeting NA.

ID	Sequence	Strain	Assay	Activity (mM)	Ref
Peptide P	PGEKGPSGEAGTAGPPGTPGPQGL	A/Puerto Rico/8/1934 (H1N1)	Cytopathic Effect Reduction	471 ± 12 g/mL.	[152]
P2	errKPAQP	A/Puerto Rico/8/1934 (H1N1)	Cytopathic effect inhibition	2.26 ± 0.40	[153]
		A/Vietnam/1203/2004 (H5N1)		1.46 ± 0.12	
IntPep	ELVDPVVAAGAVVTSSGIVFFS		Structural studies		[156]

## 5. Peptides Targeting PB1

Another target of anti-Influenza peptides is the Ribonucleoprotein (RNP) complex which comprises the nucleoprotein (NP) and the RNA-dependent trimeric RNA polymerase, which in turn includes the three P proteins (PB1, PB2, and PA) [157]. The enzyme complex binds to the 5-terminal sequence with greater affinity than the 3-terminal sequence of the vRNA [158]. The RNB complex deals with the transcription and replication of RNA segments (vRNA) in the nuclei of infected cells, generating copies of positive polarity RNA (cRNA) that form RNP complexes similar to those present in viruses [159]. The three P proteins are linked together through intramolecular interactions: the N-terminal sequences of PB1 interact with the C-terminal region of the PA subunit, while the C-terminal sequences of PB1 interact with the N-terminal region of the PB2 subunit [160]. PB1 is the protein that makes up the polymerase domain [161]. On the other hand, the PA has a series of functions, such as endonuclease and protease, and confers structural stability to the trimeric complex [162]. The PA, furthermore, contains a series of domains found in many viral species, indicating its importance in polymerase activity. The PB2 subunit has cap-binding and cap-dependent endonuclease activities [163]. However, all three P proteins are essential for transcriptional and cap-dependent activity [164].

Pharmacological study, in recent years, has focused on inhibiting the interaction of PA-PB1 because it is essential for viral replication [165]. Through an amplified luminescent proximity test (AlphaScreen), a peptide (PB1-0) was tested, consisting of the first fourteen amino acids of the N-terminus of PB1. The PB1-0 peptide has given excellent results in blocking the PA-PB1 interaction, in particular in the C-terminal domain of the PR8 (H1N1). The same results were confirmed by surface plasmon resonance (SPR) experiments [166]. Further AlphaScreen tests were performed on the truncated peptides deriving from PB1-0 to identify the smallest region capable of inhibiting the PA-PB1 interaction. The peptides deriving from PB-0 are shown in Table 5. PB1-10 was obtained with the elimination of some amino acids from the N-terminus and C-terminus. With the addition of two substituents, the authors reached PB1-11, which proved to be the most active of all [166]. The CPA-PB1-11 complex was then crystallized, showing symmetry with the PA-PB1 complex [166]. However, further chemical modifications remain necessary to improve metabolic stability and in vivo activity.

In recent years, a great deal of attention has been given to natural substances such as azurin. The latter is a copper-containing protein, produced by the bacterium I, and is structurally similar to an immunoglobulin [167]. In vivo and in vitro, azurin has already shown antitumor [168], antiviral (HIV-1), and antimalarial activity through its active site, which is located in a region consisting of twenty-eight amino acids known as p28 [169]. Docking studies showed that the p28 region forms strong and stable interactions with the C-terminus of PB1 (C-PB1) and other proteins of H1N1 IAV [170]. This makes it a great candidate for a flu drug. Further efforts are needed to evaluate its ability to neutralize the type A Influenza virus.

**Table 5 ijms-23-11433-t005:** Most representative peptides targeting PB1.

ID	Sequence	Strain	Assay	Activity (mM)	Ref
PB1-11	DYNPYLLFLK	H1N1	AlphaScreen	13 ± 1	[166]
p28	LSTAADMQGVVTDGMASGLDKDYLKPDD	H1N1	Docking studies	22 bonds	[170]
FluAPep 1	SRARIDARI	H1N1	Docking studies	binding affinity (score = −246.155)	[171]

## 6. Other

The M2 protein plays an important role in the Influenza virus: it allows the viral particle to escape the cellular defense system, stopping the vesicles (endosomes) capable of enclosing the virus and degrading it in the host cell. These vesicles are blocked by M2, which favors the escape of the virus inside the cell, where the genome replication takes place, necessary for the spread of the infection [172]. In fact, it is a transmembrane ion channel found on the virus’s external envelope and regulates the pH in the transport vesicles. The acidification pH weakens the interaction between M1 and RNP complexes and the membrane fusion allows the release of the uncoated RNPs into the cytosol.

Furthermore, the internal acidification of the virion allows the membrane fusion mediated by hemagglutinin, which in turn results in the removal of the flu nucleocapsid lining and the release of the viral RNP in the infected cell [173].

Regarding M2 as a target, great importance is being given to vaccines based on the extracellular domain of the M2 protein of the Influenza A virus [174]. The M2 protein is highly conserved, and an M2-based vaccine could be universal and develop antibodies against different subtypes of the virus [175]. The classic anti-Influenza vaccines induce immunity against proteins such as neuraminidases and hemagglutinin, especially in hypervariable regions. These regions allow the virus to escape the antibodies developed with recent vaccines. This explains why it is necessary to develop new vaccines against seasonal flu every year [176]. The universal vaccine, with a broad spectrum of action and long-lasting, could counteract the onset of pandemics.

Interestingly, Drin’s group identified the cytoplasmic domain of the M2 protein as a peptide with an alpha helix structure consisting of 18 amino acids [177]. This peptide is called M2 AH and has been seen to insert itself into the lipid bilayer of the membrane, altering it [178]. Starting from M2 AH, several peptides have been synthesized. Various assays were performed on M2 AH derivatives, such as the plaque reduction assay on MDCK cells and the MTT cytotoxicity test [179]. M2 MH proved to be the most active of all. The same was tested on different strains and maintained a robust anti-flu activity in all cases. The MTT assay shows that M2 MH is the least toxic and the most selective. This study, therefore, demonstrates that M2 MH caused the deformation of the viral membrane, neutralizing the infectivity of the Influenza virus. Hence, M2 MH has been shown to have broad-spectrum antiviral activity against several Influenza virus strains. These molecules provide several advantages over the classic antiviral peptides: the broad spectrum of action and reduced onset of resistance [76].

## 7. Conclusions

Influenza represents one of the most important plagues worldwide. Even though some antivirals are available, there is a continuing need for new anti-Influenza therapy using novel targets and creative strategies due to the alarming increase of virus strains resistance. Peptides represent a valid alternative to small molecules since they are characterized by low toxicity and a broad spectrum of action, however they have a poor pharmacokinetic profile and, especially, a low bioavailability. Disadvantages, however, have already been partially overcome with the new technologies. As described in this paper, most of the peptides studied with anti-Influenza activity target hemagglutinin. Nevertheless, therapeutic applications have not yet occurred for most of the peptides identified. Despite this, the research on antiviral peptides continues, as they represent a valid tool for a new antiviral therapy.

## Figures and Tables

**Figure 1 ijms-23-11433-f001:**
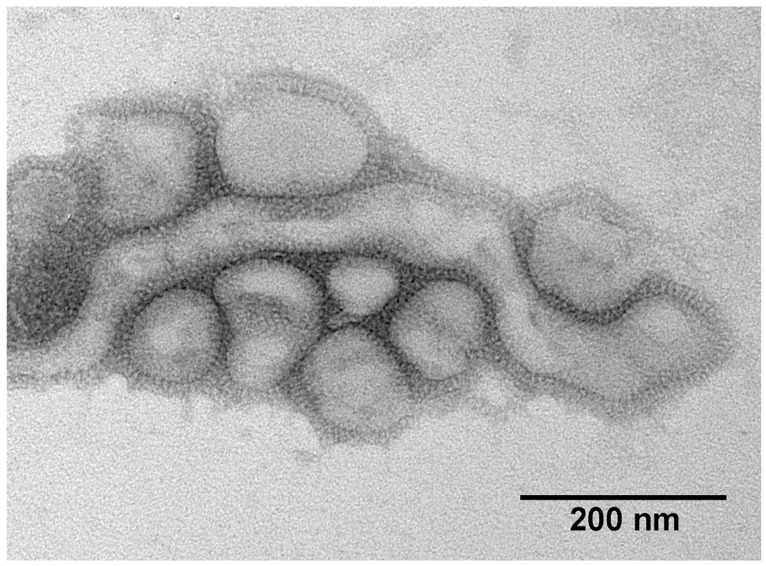
Electron micrograph of IAV particles.

**Figure 2 ijms-23-11433-f002:**
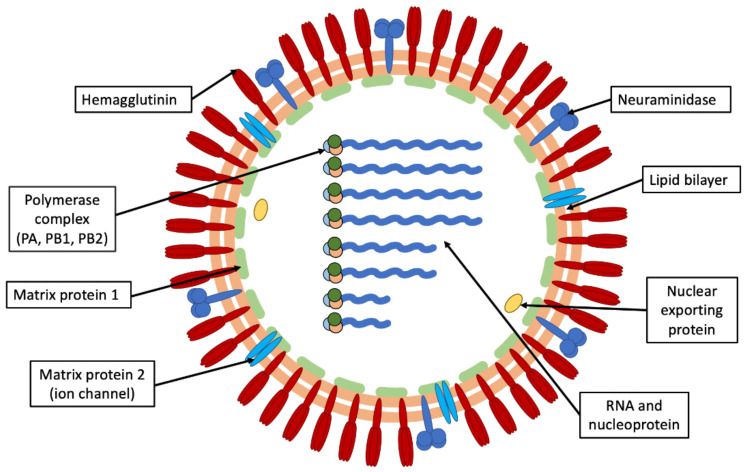
Schematic diagram of IAV particle showing viral components.

**Figure 3 ijms-23-11433-f003:**
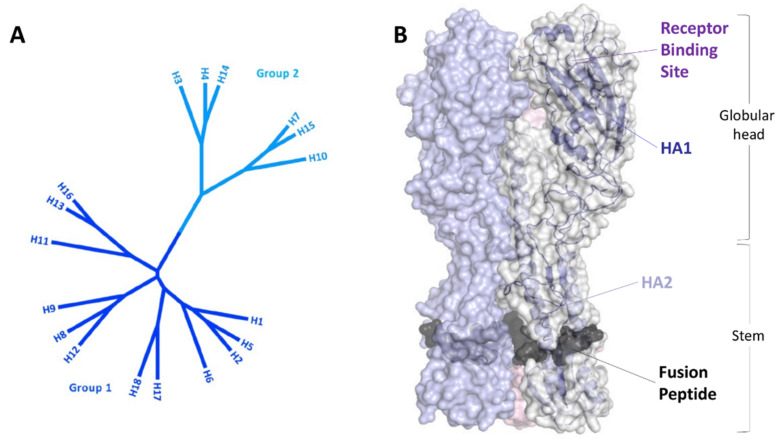
(**A**) Phylogenetic tree and (**B**) structure of hemagglutinin.

**Table 1 ijms-23-11433-t001:** IAV genes and proteins they encode.

RNA Segments	Genes	Proteins
1	PB2	Basic polymerase 2 *
2	PB1	Basic polymerase 1 *
	PB1-F2	PB1-F2 protein *
3	PA	Acidic polymerase
4	HA	Hemagglutinin
5	NP	Nucleoprotein
6	NA	Neuraminidase
7	M1	Matrix 1 protein
	M2	Matrix 2 protein
8	NS1	Non-structural protein 1
	NS2/NEP	Nuclear export protein

* RNA-dependent RNA polymerase subunits.

**Table 2 ijms-23-11433-t002:** Most representative peptides interfering with the sialic acid binding to HA.

ID	Sequence	Strain	Assay	Activity (μM)	Ref
EB	RRKK AAVA LLPA VLLA LLAP	A/Puerto Rico/8 (H1N1)	Plaque reduction	7.0 ± 2.1	[92]
B10^NP^	RRKK ______L_A VLLA LLA	A/Puerto Rico/8 (H1N1)	Plaque reduction	5.0 ± 4.0	[92]
B7^NP^	RRKK __VA LL _A VLLA LLA	A/Puerto Rico/8 (H1N1)	Plaque reduction	0.3 ± 0.2	[92]
EB extract	RRKK AAVA LLPA VLLA LLAP DDDD KHHH HHH	A(H1N1) pdm	Viral inhibition replication	0.00202 ± 0.001027	[96]
C18-s2	C_17_H_35_CO-ARLPRTMVHPKPAQP-NH_2_	A/Puerto Rico/8 (H1N1)	Plaque reduction	11	[98]
C18-s2(1–5)	C_17_H_35_CO-ARLPR-NH_2_	A/Puerto Rico/8 (H1N1)	Plaque reduction	1.9	[98]
4	Dumbbell(1)6-ARLPR	A/Puerto Rico/8 (H1N1)	Plaque reduction	0.72	[100]
7-1 peptide	C_17_H_35_CO-LVRPLAL	A/Aichi/2/68 (H3N2)	Plaque reduction	6.4	[102]
P1	SKHSSLDCVLRP	A/Parma/24/09 (H1N1)	Neutralization	3.1 ± 0.12	[110]
P2	AGDDQGLDKCVPNSKEK	A/Parma/24/09 (H1N1)	Neutralization	3.4 ± 0.14	[110]
P3	NGESTADWAKN	A/Parma/24/09 (H1N1)	Neutralization	0.05 ± 0.0014	[110]
14	VLRP	A/Parma/24/09 (H1N1)	Neutralization	1 ± 0.05	[112]
15	SLDC	A/Parma/24/09 (H1N1)	Neutralization	4.6 ± 0.05	[112]
17	SKHS	A/Parma/24/09 (H1N1)	Neutralization	0.048 ± 0.0012	[112]
4	SAHS	A/Parma/24/09 (H1N1)	Neutralization	0.0004 ± 0.00003	[113]
PeB	ARDFYDYDVFYYAMD	A/Aichi/2/68 X31 (H3N2)	Infection inhibition	32 ± 5	[114]
PeB^GF^	ARDFYGYDVFFYAMD	A/Aichi/2/68 X31 (H3N2)	Infection inhibition	25 ± 6	[114]
C18-PeBGFa	C_17_H_35_CO-ARDFYGYDVFFYAMD	A/Aichi/2/68 X31 (H3N2)	Infection inhibition	5.9	[115]
4b	dPG_340_PeB_9 Ligand_	A/Aichi/2/68 X31 (H3N2)	Infection inhibition	0.3 ± 0.1	[116]
4b	dPG_340_PeB_9 Nanoparticle_	A/Aichi/2/68 X31 (H3N2)	Infection inhibition	0.0006 ± 0.0003	[116]
L-P1	NDFRSKT	A/chicken/Iran/16/2000 (H9N2)	In ovo antiviral activity	48	[117]
C-P1	CNDFRSKTC	A/chicken/Iran/16/2000 (H9N2)	In ovo antiviral activity	71	[118]
P1	LSRMPK	A/chicken/Tunisia/12/2010 (H9N2)	In ovo antiviral activity	870	[119]
P2	FAPRWR	A/chicken/Tunisia/12/2010 (H9N2)	In ovo antiviral activity	620	[119]
iHA-100	Ac-WTGDFFSSHYTVPRC	H5 HA	Surface Plasmon Resonance	0.0015	[120]
PVF-tet	(MA-RRPVNHF-AU)4-3Lys	A/Puerto Rico/8 (H1N1)	Infection inhibition	1.4	[121]

**Table 3 ijms-23-11433-t003:** Most representative peptides interfering with the fusogenic activity of HA.

ID	Sequence	Strain	Assay	Activity (μM)	Ref
	FHRKKGRGKHK	A/Hufang/7/1999 (H1N1)	Neutralization	1 log unit inhibitory activity	[123]
*HA-FP-1*	GLFGAIAGFI**K**NGW**K**GMI**K**G	A/Puerto Rico/8/34 (H1N1)	Cytopathic Effect inhibition	9.61 (µg/mL)	[124]
		A/Aichi/2/68 (H3N2)		5.90 (µg/mL)	
*C3LB-HA*	Residues 270–285 of the HA1 C-ter	A/Puerto Rico/916/34 (H1N1)	Cytopathic Effect inhibition	27.21	[125]
		A (H1N1)pdm2009		26.45	
*P155–185-Chol*	GTYDHDVYRDEALNNRFQIKGVELKSGYKDWGSGSG-C(PEG4-Chol)NH2	A/Hong Kong/8/68 (H3N2)	Plaque reduction	0.4	[126]
*IIQ*	AcIEEIQKKIEEIQKKIEEIQKKIEEIQKKIEEIQKKβAKC16	A/Puerto Rico/8/34 (H1N1)	Cytopathic Effect reduction	1.73	[128]
*HB36.6*	97 aa sequence	A/California/2009 (H1N1)	Cytopathic Effect reduction	0.18 µg/mL	[130]
		A/Puerto Rico/1934 (H1N1)		0.58 µg/mL	
		A/New Caledonia/1999 (H1N1)		1.26 µg/mL	
		A/Hong Kong/2003 (H5N1)		12 µg/mL	
*P7*	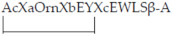	A/California/07/2009 (H1N1) A/New Caledonia/20/1999 (H1N1) A/Vietnam/1203/2004 (H5N1)	AlphaLisa competition	0.03–0.07	[131]
*FP4*	RRKKWLVFFVIFYFFR	A/Winsconsin/33 (H1N1)	Plaque reduction	0.00004	[132]
*Urumin*	IPLRGAFINGRWDSQCHRFSNGAIACA	A/Puerto Rico/8/34 (H1N1)	Plaque reduction	3.4	[133]
	C12-OOWO	A/Puerto Rico/8/34 (H1N1)	Virus titer reduction	8.48 ± 0.74 (mg/L)	[134]
	C12-KKWK	A/Puerto Rico/8/34 (H1N1)	Virus titer reduction	7.30 ± 1.57 (mg/L)	[134]
*C20-Jp-Hp*	C20-ARLPRKKWK	A/Puerto Rico/8/34 (H1N1)	Cytopathic Effect inhibition	0.53 ± 0.25	[135]

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
