# Peer review of "Antiviral Peptides as Anti-Influenza Agents"

_ijms, 2022, doi:10.3390/ijms231911433_

Round 1

Reviewer 1 Report

It is a good and comprehensive review on anti-Influenza peptides. However, I have several remarks:

1) Association of HA and NA with lipid rafts is still under debates, because there are several studies (e.g., Wilson, et al., Biophys. J., 2015) indicating that it is not true. Thus, I suggest not to say so strictly that HA and NA are anchored in lipid rafts.

2) "Inside the endosome, the protons enter the virions through the M2 channels creating an acidic internal environment [39] which, by inducing a significant conformational change in HA that exposes the fusion peptide, leads to the fusion between the viral envelope and the endosome membrane and, therefore, to the release of viral ribonucleoproteins (vRNPs) in the cytoplasm [40]" - Conformational changes of HA occur outside the virion, not inside, and they are not connected with the action of M2 channels. Please, change the phrase. Also, it would be better to add several phrases about the membrane fusion activity of HA, formation of the small fusion pore and its further expansion upon disintegration of the viral envelope.

3) Next - release of the viral RNP requires disintegration of the M1 scaffold. This fact was not mentioned, however, there are a plenty of studies regarding this fact.

4) In the Section 6 description of the function of the M2 protein is not correct, because its activity regulates disintegration of the M1 scaffold after the formation of the small fusion pore by HA molecules themselves.

Reviewer 2 Report

1. Page 2, Line 7. The origins of those pandemics were described as "presumed" in the cited reference. Using "believed" is not accurate. 

2. It may not be good under the section “Flu and Influenza Virus”. It will be better to have a new section, “Treatment of Flu”.  

In my opinion, the introduction part should include (1) a brief introduction about flu and influenza virus (which you have, the influence, and some details of its biological properties); (2) a brief introduction of some current treatments (vaccines and small- molecule drugs) and their main drawbacks, which helps to show that peptides would be promising agents to overcome them. 

I would encourage you to combine 1.1, 1.2, and 1.3, and have a new 1.2 for the treatments. 

3. Please also update your citation number by their appearance order.  For example, Page 5, Line 51. 
